# Predicting the Healing of Lower Extremity Fractures Using Wearable Ground Reaction Force Sensors and Machine Learning

**DOI:** 10.3390/s24165321

**Published:** 2024-08-17

**Authors:** Kylee North, Grange Simpson, Walt Geiger, Amy Cizik, David Rothberg, Robert Hitchcock

**Affiliations:** 1Department of Biomedical Engineering, University of Utah, Salt Lake City, UT 84112, USA; grange.simpson@utah.edu (G.S.); u1267047@utah.edu (W.G.);; 2Department of Orthopaedics, University of Utah, Salt Lake City, UT 84112, USA

**Keywords:** wearable sensors, machine learning, ankle fracture, tibial fracture, lower limb rehabilitation

## Abstract

Lower extremity fractures pose challenges due to prolonged healing times and limited assessment methods. Integrating wearable sensors with machine learning can help overcome these challenges by providing objective assessment and predicting fracture healing. In this retrospective study, data from a gait monitoring insole on 25 patients with closed lower extremity fractures were analyzed. Continuous underfoot loading data were processed to isolate steps, extract metrics, and feed them into three white-box machine learning models. Decision tree and Lasso regression aided feature selection, while a logistic regression classifier predicted days until fracture healing within a 30-day range. Evaluations via 10-fold cross-validation and leave-one-out validation yielded stable metrics, with the model achieving a mean accuracy, precision, recall, and F1-score of approximately 76%. Feature selection revealed the importance of underfoot loading distribution patterns, particularly on the medial surface. Our research facilitates data-driven decisions, enabling early complication detection, potentially shortening recovery times, and offering accurate rehabilitation timeline predictions.

## 1. Introduction

Lower extremity fractures (LEFs) are a global health concern, affecting individuals of all demographics [1,2,3]. The recovery process after these fractures presents challenges for both patients and healthcare providers due to their prolonged rehabilitation times, high complication rates, and negative impacts on activities of daily living and the ability to work [4]. Current methods for assessing rehabilitation progress rely on intermittent evaluations through non-instrumented gait and range of motion observations, patient-reported surveys, and X-rays [4,5]. However, these assessments provide limited insights into the healing progression of LEFs. Non-instrumented gait and range of motion observations lack objectivity and demonstrate low interrater reliability and responsiveness [6]. An X-ray can offer insight into callus formation and fracture mineralization but cannot capture the healing progress of secondary soft tissue injuries or limb functionality [7]. Patient-reported surveys, designed for stable health status measurement, lack sensitivity to changes and are influenced by the patient’s cognitive or emotional state [8]. Moreover, current clinical assessments lack data volume as they are only collected during clinic visits scheduled 4–6 weeks apart. These long assessment intervals create a disconnect between patients and clinicians, contribute to psychosocial distress during rehabilitation, hinder the timely detection of complications, and offer little in terms of shortening the rehabilitation time [9,10].

Wearable sensors offer a practical solution to improve the volume and quality of insights into fracture healing progression [11,12]. Wearable sensors can monitor patients in real-world settings [13], provide continuous data at a higher volume [14], and capture objective physiologic metrics [15]. Wearable sensors can be tailored to focus on specific physical phenomena relevant to LEF rehabilitation, such as gait quality. Monitoring gait has proven to be a valuable approach for gaining objective insights into the healing progression of lower extremity injuries [16,17].

Human gait involves the coordinated action of multiple interrelated body parts, described as the lower limb kinetic chain [18]. The vertical ground reaction forces (GRFs) experienced at the bottom of the foot represent the culmination of this intricate chain [19]. When continuously sampled, GRF data becomes a GRF curve. Disruptions or abnormalities with any component of the lower limb kinetic chain will result in detectable changes in GRF curves [20]. The high sensitivity of GRF curves to lower limb function makes GRF curves a robust metric for assessing gait quality during LEF rehabilitation. Specific alterations in GRF curves can signal common complications following LEFs, including antalgic gait, varus angulation, delayed union, and non-union [21,22,23].

The impact of wearable sensors is further enhanced by the use of machine learning (ML), which enables the exploration of large, complex datasets generated by wearable sensors. ML empowers the discovery of nuanced or concealed trends and patterns that were previously unidentifiable [24,25]. Additionally, “white-box” ML algorithms provide insights into the influence of specific inputs on the final output, increasing understanding of the observed trends [26].

In this study, we test the hypothesis that a wearable GRF sensor paired with ML algorithms can objectively assess LEF healing, serving as a vital sign for these injuries. In this retrospective study, we leverage existing datasets from patients undergoing LEF rehabilitation with white-box ML algorithms. This approach will predict when the patient will heal while simultaneously extracting valuable information regarding gait patterns and their association with healing outcomes. The goal of this research is to advance LEF rehabilitation assessments by increasing the data volume and insights obtained from these assessments. Enhancing rehabilitation assessments to improve data volume and insights will lead to data-informed therapeutic decisions, early detection of complications, and improved patient engagement. Ultimately, this approach can lead to objective therapeutic decisions, reduced recovery times, and improved outcomes.

## 2. Materials and Methods

### 2.1. Study Design

The basic study design is to feed gait quality metrics from the GRF curves of a single step into a logistic regression classifier and predict the number of days until the patient’s fracture will heal.

This is a retrospective study of an IRB-approved 3-year prospective clinical observational trial using a gait monitoring insole to continuously record GRF curves from a walking boot cast. A total of 42 patients were enrolled with injury criteria as follows: a closed tibial or bimalleolar ankle fracture (AO/OTA classification 42–44), 18 years of age or older, English-speaking, weigh between 45.3 and 113.4 kg, live within 161 km of the hospital, and not have multiple injuries. A study team member performed recruitment before surgery, and the treating clinicians were blinded to patient participation. Informed consent was obtained from all subjects involved in this study. Patients received the instrumented boot cast at their two-week post-operative appointment. The date of bone healing was determined from the X-rays that were taken periodically during this study per standard of care. Additional details regarding the gait monitoring insole and the clinical protocol have been previously reported [27,28,29].

### 2.2. Data Reduction

#### 2.2.1. Step Extraction and Discretization

The gait monitoring insole consisted of three load sensors positioned under the heel, medial forefoot, and lateral forefoot areas. The insoles were customized to fit the patients’ instep length, ensuring proper placement of the forefoot sensors. Each sensor was sampled simultaneously at 16 Hz [30]. We developed a customized step-isolation algorithm using a wavelet transform and signal reconstruction. For the isolated step to be extracted for further analysis, the step had to have a cadence between 55 and 160 steps/minute [31,32], go to swing phase on both sides, and be free of sensor errors. The extracted steps were separated into the three sensor waveforms: heel, medial, and lateral. Three calculated waveforms were derived from the three sensor waveforms: total load, forefoot (sum of medial lateral sensors), and the difference between heel and forefoot (Diff_HF). In total, 129 metrics were calculated for each step from the six different waveforms. Metrics could be calculated with the time domain in one of two configurations, either as the raw signal or normalized to 100 data points per gait cycle using an interpolation method. The magnitude of the waveform also had two configurations: normalized to the patient’s weight or the non-normalized signal in kilograms. See Table 1 for details on the calculated metrics and configurations for each waveform. See also Figure 1 for an overview of the data collection.

All calculated metrics for each step were compiled into a dataframe and integrated with the patients’ clinical data, which included BMI, weight, gender, injury energy rating, OTA fracture classifications of the location and complication level, and age at injury. To categorize the data, the recorded date was subtracted from the date of fracture healing, yielding the ‘days to healed’ (DH) metric. Since X-rays were used to determine the healing date and were taken at approximately 4-week intervals, a 30-day range was used for separating the DH metric into five bins and storing it in the DH feature. A DH0 metric indicates that the fracture would heal between 0 and 30 days, a DH1 metric is between 30 and 60 days, DH2 is between 60 and 90 days, DH3 is between 90 and 120 days, and DH4 is 120 days or longer.

#### 2.2.2. Random Sampling and Feature Selection

To ensure equal representation of each patient in each DH bin and minimize bias, a random sampling function was developed to compensate for the variable number of steps taken by patients during the healing process. The function iterated through each patient, partitioned their data into different DH bins, randomly extracted 500 samples from each bin, and saved them to a dataframe named Uniformed_Sampled_Steps. The remaining unsampled data were stored in a separate dataframe.

We employed an ensemble ML approach to extract features that utilized decision trees and lasso regressions. This combination of lasso regression and decision tree helped ensure the inclusion of non-linearly correlated data and facilitated optimal feature selection. Metrics such as patient BMI and weight were also excluded from the training dataframe, so the model would not be trained on the patient.

The dataset was randomly split into an 80/20 train–test split. We set the decision tree classifier’s maximum depth to 6 and used entropy as the criterion. Based on their feature importance values, we extracted the top 15 features. The data were scaled in the case of lasso regression, and L1 regularization was employed to penalize less relevant features. The regularization parameter was finetuned using cross-validation, and the top 15 features were exported.

The variance inflation factor (VIF) for each selected feature in the different lists was calculated to test for multicollinearities. We systematically replaced features from the decision tree feature list that exceeded the VIF cutoff value of 10 [33] with those from the lasso feature list. This systematic replacement was performed until we obtained a final feature selection list that satisfied the VIF criterion and performed well in logistic regression. The Uniformed_Sampled_Steps dataframe was reduced to the feature selection list and the DH bin classifier. The Pearson correlation coefficient was calculated between the numeric DH bin classification and the selected features, with a *p*-value of < 0.05 determining significance. An overview of the ML methods is shown in Figure 2.

### 2.3. Logistic Regression Classifier

The reduced Uniformed_Sampled_Steps dataframe was split into an 80/20 train–test ratio. To normalize data and prevent information leaks between the training and test datasets, the mean and standard deviation were calculated from just the training dataset and saved to normalize all training and input test data. 

To improve the accuracy of this multiclassification task, a custom function was created that combined the results of five separate logistic regressions. The function was developed in Python and utilized the sklearn logistic regression library. The five logistic regressions were trained with the same insole and patient metrics; what changed between the functions was the DH classifier. Five separate classification lists were created from the original DH classifier, which was trained on data targeting each DH bin. The classification of the DH bin of interest remained the same. The data not in the DH bins were assigned classifications as either above or below the bin of interest based on their numeric value (see Figure 1). Each logistic regression was named after the targeted DH bin. Parameter optimization was performed using sklearn’s GridSearchCV across all five logistic regressions and 25 patients, with balanced accuracy as the metric for each logistic regression’s performance. Optimal parameters were saved for each patient and logistic regression, and in the event of a difference in optimal parameters between patients, the parameter with the majority across all patients was saved for each logistic regression. When running the optimized models, the probability of input data belonging to the targeted DH bin was extracted from all five logistic regressions. The highest DH bin probability was the predicted class produced by the algorithm. If the probability was less than 55%, that row of data was labeled to be omitted from the final calculations. The importance score for each feature from all five logistic regressions was exported. See Figure 2 for an overview of the ML methods.

A 10-fold cross-validation (resampling the training and test data 10 times) and a “leave one patient out” cross-validation were performed to ensure the model’s accuracy and generalizability were robust [34]. For all cross-validations, the accuracy, weighted precision, recall, and F1-scores were exported.

After the model was validated, all the patient data was run through the model for a final time. The predicted classifier was merged into the original table for further examination. Two patient cases were examined more closely, one with a high volume of steps and healed normally, the other with an asymptomatic delayed union.

### 2.4. Examining Trends in Selected Features

The logistic regression outputs an importance score for each feature, which ranks the importance of that feature for the classification [35]. We, therefore, looked at the trends between the DH values for each selected feature and the importance scores associated with those features. The selected features were divided into two groups: categorical features and continuous features.

To assess the normality of the feature values within each DH bin, we used the Jarque–Bera test. In cases where the test indicated a non-normal distribution, we examined the statistical differences between groups using the Kruskal–Wallis test. If the Kruskal–Wallis test revealed significant differences among the groups, we performed Dunn’s comparison test with Sidak post hoc correction on the data to identify specific group differences. 

## 3. Results

### 3.1. Study Design

Data were collected from 25 patients (16 female and 9 male; 77.7 + 15.7 kg, 26.8 + 5.8 BMI, and 37.6 + 15.3 y/o). Of the original 42 patients enrolled, 7 were lost to follow-up or removed from this study, and 10 had insole errors. In total, 908 days of data were collected between all the patients, averaging 36.3 days of data per patient.

### 3.2. Data Reduction—Step Extraction and Discretization 

A total of 390,970 steps were extracted; each step had 138 features between the step metrics, patient data, and categorical lists. The ‘days to healed’ ranged from 5 to 519 days. Three patients had complications during this study, resulting in longer ‘days to healed’ for all their steps. Not all patients were represented in each DH bin since some did not wear the boot cast with the insole before reaching the lower DH bins, which approached healing. In other cases, patients healed faster than others and did not have data in the DH bins that were further from healed. Patients ranged from having steps classified between 1 and 3 of the 5 DH bins. 

### 3.3. Data Reduction—Random Sampling and Feature Selection 

The dataframe of randomly sampled patient step metrics contained 19,422 steps: 4822 in DH0, 7434 in DH1, 4331 in DH2, 1289 in DH3, and 1546 in DH4; see bar chart in Figure 2. The final feature selection list had the following features; OTA fracture location, OTA fracture complication rating, the mean of the raw Diff_HF waveform, the location of the maximum of the Diff_HF waveform, the ratio of the heel to the medial sensor, the ratio of the lateral to the medial sensor, the variance of the total waveform, gender, the mean of the raw medial sensor, and the variance of the raw medial sensor (var M), and the day count of sensor recording (Day). Table 2 lists the selected features, their VIF values, and the Pearson correlation coefficient to the DH bin classification (r to DH).

### 3.4. Logistic Regression Classifier

The logistic regression function had an accuracy of 76%. Filtering out steps, or rows of data, with a predictive probability of less than 55% results in a 23% step reduction. The results of the 10-fold cross-validation and leave one patient out cross-validation on precision, recall, F1-score, and accuracy are shown in Table 3. The patient case study showed that patient 25 progressed through the healing bins as expected, going from DH2 (60–90 days to healed), through DH1 (30–60 days to healed) to DH0 (0–30 days to healed). Patient 6 started in DH1 and digressed to DH4 and DH5 (90 + days to healed) (see Figure 3).

### 3.5. Examining Trends in Selected Features 

The mean of the medial waveform had the highest importance scores of all the gait features. The variance of the medial waveform, the max heel-to-max medial ratio, and the mean of the Diff_HF waveform also had large importance scores. We examined the trends between the mean of the medial waveform, the heel-to-medial ratio, and the mean of the Diff_HF waveform. As DH4 and DH3, which represent steps that were at least 90 days from healed, were under-sampled, their data may not be representative or reliable enough to report. Therefore, we only report the results from trends within DH3–DH0 bins, representing data points between 0 and 90 ‘days to healed.’ More data from patients in later ‘days to healed’ ranges is needed to draw more reliable conclusions. The general trend was that the mean of the medial waveform increased as patients approached healing. There was a significant difference between each group according to the Dunns test with a Sidak post hoc correction (Dunn–Sikad). The max heel to max medial ratio had the general trend of the medians increasing while the means decreased as the patient approached healing. The variation in values was greater for the DH2 bin, which is 60–90 ‘days to healed’. There was a significant difference between all groups according to the Dunn–Sidak test. The mean value of the Diff_HF waveform increased as the patient approached healing; however, the increase was not significant between DH2 and DH1. Figure 4 displays the waveform morphology for the medial, heel, and Diff_HF waveform for patient 25 to give a visual representation of the changes in the GRF. Boxplots for the three metrics and a bar chart displaying importance scores from the different logistic regressions are also shown. 

## 4. Discussion

For this study, we hypothesized that a wearable GRF monitoring device combined with ML could predict when a patient will heal from an LEF fracture, thus serving as an objective tool for monitoring healing progression. The ability to predict fracture healing based on objective physiological data would enable timely interventions and escalation of care and provide patients and clinicians with a tangible endpoint of care. To test this hypothesis, we employed a unique approach that randomly isolates steps from GRF curves recorded by a gait monitoring insole at different stages of patient LEF healing. Steps were discretized into various metrics calculated from the GRF curves. These metrics were then fed into a logistic regression classifier, enabling patients to be classified based on the number of days until their fracture was expected to heal. We evaluated the performance of our model using 10-fold cross-validation and leave-one-out validation and assessed key metrics, including recall, precision, F1-score, and accuracy. Our model demonstrated strong performance across all metrics, ranging between 74 and 79% (see Table 3), providing robust support for our hypothesis. 

Among the white-box models used to gain insight into gait alteration during recovery, the logistic regression classifier emerged as the best-performing algorithm for gait data [35,36,37]. Before training the logistic regression, we employed two white-box feature selection algorithms, decision tree and lasso regression. This feature selection process served two purposes: identifying the gait components that provided the most valuable insights into patient healing and reducing the initial set of 139 metrics to a subset without multicollinearities. Lasso regression excels at producing a dataset with uncorrelated features but does not do well with non-linear data [38]. A decision tree does very well with non-linear data but is less effective than lasso regression at producing uncorrelated datasets [39]. By leveraging the strengths of each algorithm in an ensemble approach, the resulting feature selection outputs resulted in a set of uncorrelated features from a non-linear dataset. 

The selected features highlighted the importance of underfoot loading distribution patterns. The pattern that emerged from all the distribution-loading-related metrics demonstrated that loading was predominantly towards the forefoot in the early stages of rehabilitation, which aligns with the more toe-touch gait during early weight bearing. As healing progressed, the peak loading shifted to the heel to align more with a normal, healthy gait that starts with a heel strike. This change in loading patterns is best represented in Figure 4A in the Diff_HF waveform. The prominence of the medial sensor in multiple features, along with its high importance scores from the logistic regression, suggested that loading at the medial forefoot also played a critical role in assessing healing progression. Future investigations using wearable sensors should include measuring from multiple locations underfoot and focusing on the medial forefoot loading and the distribution between the heel and the forefoot.

Wearable sensors allow data to be collected in out-of-clinic settings, providing a more comprehensive view of patient activity [13,15]. However, continuous data collection generates large datasets that can be challenging to manage and interpret [14]. The data used for this study is one of only three datasets of continuous patient loading during fracture rehabilitation [40,41,42]. In total, there were 908 days of data collected. The extracted steps from this dataset varied widely due to significant variations in human activity throughout the day. The continuously recorded gait data encompassed different types of walking, from shuffling to consistent walking bouts, and walking on various terrains such as stairs, grass, and flat surfaces. These variations introduce complexities that make traditional data analysis methods insufficient. Machine learning, however, enables the exploration and identification of nuanced or concealed trends and patterns within these large, complex datasets [24,25].

To assess the model’s performance, we employed two validation methods: 10-fold cross-validation and leave-one-out validation. During these validations, essential performance metrics such as precision, recall, F1-score, and accuracy were examined. These metrics remained stable across the validations, with variations of less than 5% between the minimum and maximum values. This consistency indicates that despite the relatively small sample size of 25 patients, the model is robust and is expected to perform consistently with new data, demonstrating generalization in that the model is not overtrained to the specific dataset. Additionally, the reported performance values were notably high, hovering around 76%, which is comparable to the accuracy, precision, and recall values observed in other medical screening and diagnostic tools [43,44,45,46]. The experimental setup involved training a logistic regression classifier to estimate when a patient’s fracture will heal within a 30-day range based on metrics from a single step. Using a single step served multiple objectives, including testing the model’s hypothesis and eliminating the need to hand-pick steps or epochs (sections in the signal) of loading waveforms. Eliminating the need for a user to select steps or epochs as a means of data isolation reduces bias and facilitates easy integration of the model into clinical care without imposing additional requirements on patients or clinicians during data collection. Future work developing criteria for selecting steps that yield more accurate classification results would improve overall system performance. Additionally, the 30-day range used for classification had a strict cutoff, leading to some overlap in the surrounding days, ultimately impacting accuracy. Further research could mitigate this inaccuracy by refining the model setup or using other metrics that are more frequent than X-rays to determine healing.

Methods for improving accuracy, such as generating interaction terms or using neural networks [35,36], were explored, but they compromised general audience interpretability while providing minimal increases in accuracy. We strongly emphasized interpretability within our algorithm for both the inputs and outputs since a key objective of this research was to ensure effective communication of our findings to a diverse audience. For those reasons, these methods are not discussed in this paper.

The case study shown in Figure 3 demonstrates the utility of ML in predicting healing trajectory. Most patients displayed the expected behavior of moving down classification bins as healing progressed. Patient 25 is a notable example of this pattern, healing without complications on day 83 post-operatively (Figure 3). In contrast, patient 6 deviated from the expected trajectory. Initially predicted to heal within 30–60 days, at day 80 post-operatively, the logistic regression started classifying this patient as expected to heal in 90 days or more. Coincidentally, this patient had a clinical follow-up appointment on day 87 post-operatively. Despite collecting the standard of care rehabilitation assessments, no deviations in healing were detected during that appointment. Eventually, this patient was diagnosed with an asymptomatic delayed union, which took 206 days to heal. This case demonstrates the potential of wearable sensors and ML to detect and thus prevent such scenarios in order to improve patient outcomes. 

Providing patients with feedback on their expected healing time adds significant value to their clinical care, fostering a more engaged and informed approach to rehabilitation. The landmark Lower Extremity Assessment Project (LEAP) study, which spanned over seven years and involved multiple study sites, reported that immutable factors such as sex, age, and injury severity did not have the most significant impact on quality of life. Instead, the mutable factor of self-efficacy—the belief in one’s capacity to heal and act accordingly—had the most significant impact on long-term outcomes [9]. Furthermore, in a recent survey evaluating the value of patient-generated health data (PGHD) from various perspectives, including patients, care partners, clinicians, and hospital administrators, the theme of PGHD supporting care decisions and improving patient-provider communication emerged as the most dominant [47]. Thus, providing patients with a more defined healing end period based on objective monitoring and data-driven insights can enhance their experience, instill confidence in their rehabilitation progress, and actively engage them in their care.

Overall, our findings show that combining wearable GRF sensors with ML can objectively monitor healing progression in lower extremity fracture rehabilitation. ML was able to extract practical and understandable insights from the large and complex data set produced by the continuous GRF sensor. These insights not only enable timely interventions and informed therapeutic strategies but also enhance communication between clinicians and patients. By delivering practical and objective information about the healing process, our research promotes improved patient self-efficacy and advocates for a more data-driven approach to lower extremity fracture care.

## 5. Conclusions

This study demonstrates the transformative potential of combining wearable sensors with ML to enhance healthcare delivery, particularly in the context of LEF rehabilitation. Wearable sensors facilitate the collection of high-quality, objective physiologic data in large volumes, while ML enables the exploration of these complex datasets to uncover nuanced or previously hidden trends and patterns. We paired a wearable ground reaction force sensor with ML algorithms to predict when a patient will heal from a LEF, thus serving as an objective tool for assessing LEF healing progression. Our approach involved isolating steps from ground reaction force curves recorded by a gait monitoring insole at different stages of patient healing. Metrics derived from these steps were fed into a logistic regression classifier, which then predicted the number of days until the fracture was expected to heal. The model’s performance was evaluated using 10-fold cross-validation, leave-one-out validation, and key metrics, including recall, precision, F1-score, and accuracy, demonstrating strong performance across all metrics (74–79%).

Additionally, our use of white-box ML algorithms provided valuable insights into gait trends during the healing process. Specifically, we observed that early in the healing process, there is reduced loading on the medial part of the foot, which increases as the patient approaches full recovery. There is also a shift from predominant forefoot loading in the early stages to a combination of heel and forefoot loading as healing progresses.

Providing patients with feedback on their expected healing time significantly enhances their clinical care, fostering a more engaged and informed approach to rehabilitation. Objective monitoring and data-driven insights can instill confidence in patients regarding their rehabilitation progress, actively engage them in their care, and ultimately improve their overall experience and outcomes.

## Figures and Tables

**Figure 1 sensors-24-05321-f001:**
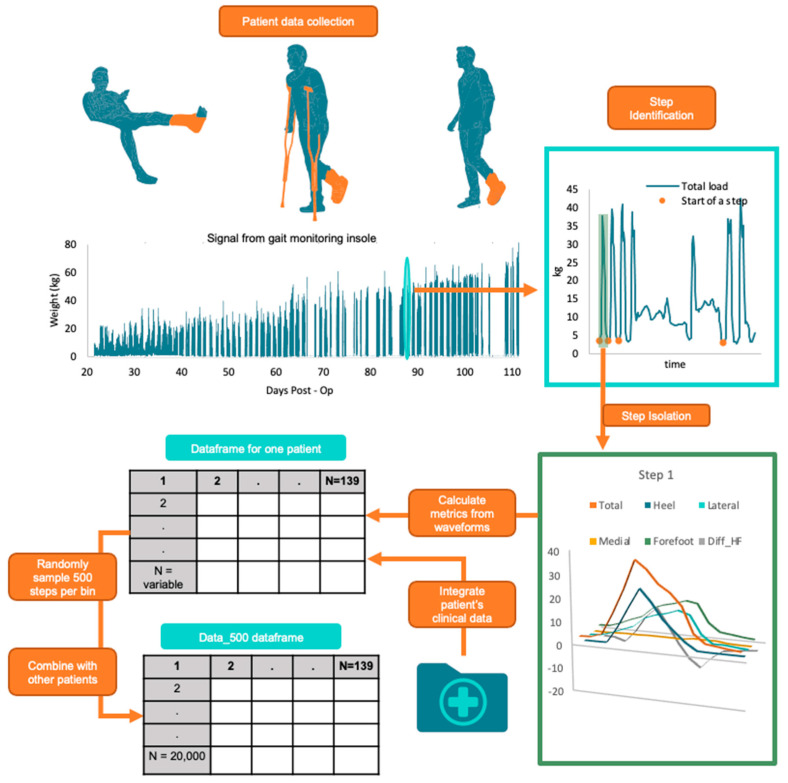
Pipeline for collecting patient gait data continuously during lower extremity fracture rehabilitation. Steps were identified using a wavelet transform, followed by isolating a single step with six distinct waveforms. Metrics derived from the isolated step were then calculated and integrated with clinical data and other patient dataframes.

**Figure 2 sensors-24-05321-f002:**
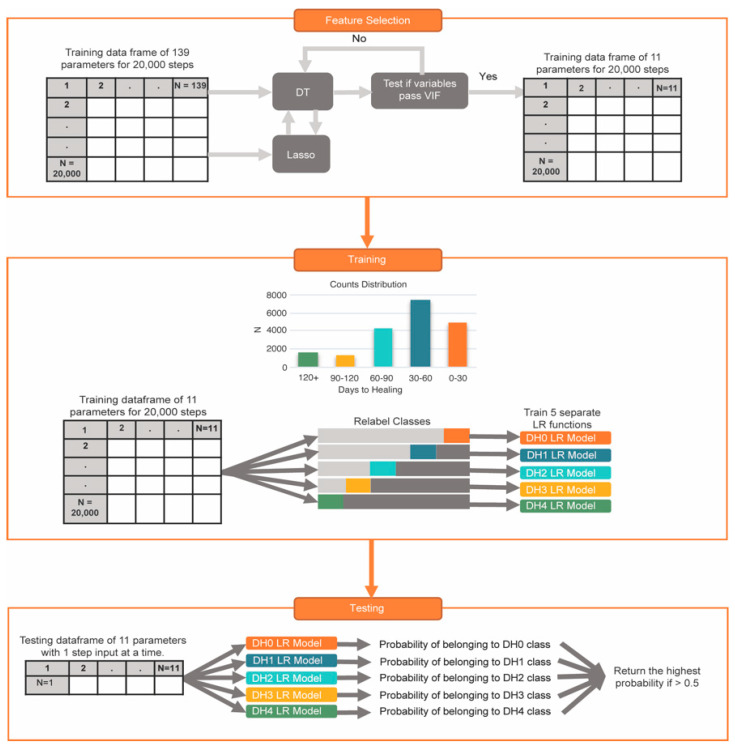
The feature selection process involved iterating through both feature lists until the variance inflation factor (VIF) was less than 10. The selected features were then used to reduce the dimensionality of the dataframe. Subsequently, five separate logistic regression models were trained using the reduced feature set. During classification, a single step was passed through all five logistic regression models, resulting in the probability of belonging to each class. The step was classified based on the highest probability obtained from the models.

**Figure 3 sensors-24-05321-f003:**
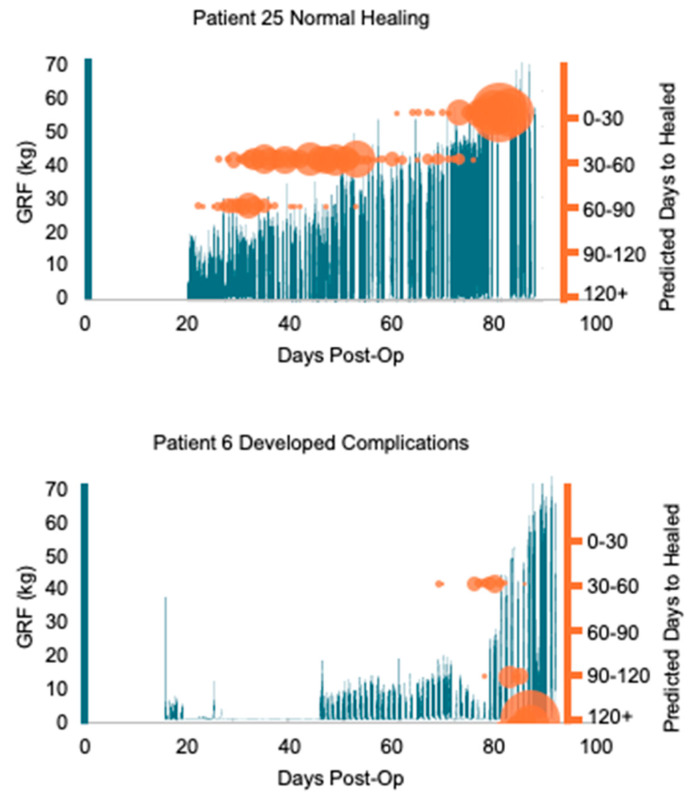
Case study showing gait monitoring GRF data and logistic regression classification function results for two patients. The GRF recorded from the gait monitoring insole is displayed for Patient 25 and Patient 6. The GRF data (in blue on left axis) for both patients is overlaid with the results from the logistic regression function classification (in orange on the right axis). The size of the orange data marker indicates how many steps were assigned that “Days to Healed” classification from the logistic regression for that day. Patient 25 demonstrates a progressive pattern through the classification bins as expected, indicating a timely healing progression. In contrast, Patient 6 shows a regression in their healing trajectory, transitioning from an initial classification of expected healing within 30–60 days to a classification of 120+ days. Patient 6 developed complications during rehabilitation.

**Figure 4 sensors-24-05321-f004:**
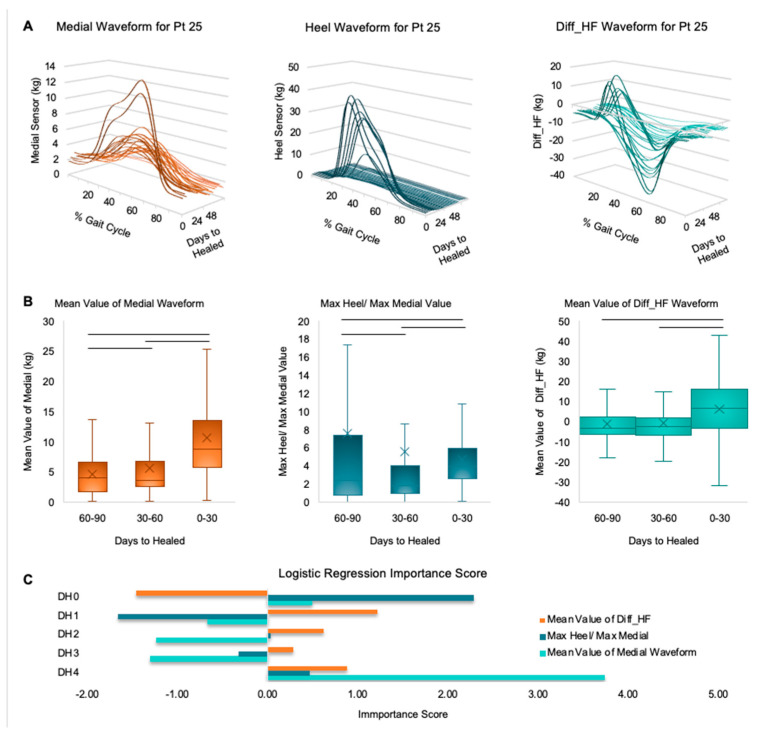
Panel (**A**) showcases the changing morphology of three waveforms throughout the healing progression. Panel (**B**) displays box plots are presented for the three most important features; the central line represents the mean, the X represents the median value, and the bars above indicate statistically significant (*p* < 0.03). Panel (**C**) provides the importance scores of the three metrics in each of the five logistic regression models.

**Table 1 sensors-24-05321-t001:** Metric configuration.

Feature	FS Model	VIF	Importance Score					
			DHA	DH4	DH3	DH2	DH1	DH0
OTA complication	Both	1.67	−3.20	4.59	−2.78	−1.76	−1.65	1.80
OTA location	Both	1.44	2.38	−5.89	2.79	1.48	0.95	−0.78
Mean of the medial sensor	DT	1.60	−1.25	3.74	−1.30	−1.23	−0.67	0.49
Day	Both	1.23	2.99	−0.37	0.25	1.24	2.32	−2.99
Variance of medial sensor	DT	2.44	1.60	−1.68	0.86	1.46	1.04	−0.96
Heel to medial ratio	Lasso	5.19	−1.84	0.46	−0.32	0.03	−1.66	2.28
Mean of Diff_HF	Lasso	1.13	1.33	0.88	0.28	0.62	1.22	−1.46
Gender	Both	2.70	−0.79	2.41	−0.63	0.00	−0.48	0.83
Variance of the total waveform	Lasso	2.86	0.20	−3.02	0.55	−0.37	−0.08	−0.13
Lateral to the medial ratio	DT	4.81	−1.00	0.14	−1.11	−2.43	0.02	−0.04
Location of the maximum on Diff_HF	Lasso	3.55	−0.29	0.01	0.04	−0.12	−0.29	0.41

Note: FS = features selection; DT = decision tree; VIF = variance inflation factor; DH = bins based on days to healed; DHA = importance score from logistic regression trained to classify all DH bins; DH4-0 = importance score from logistic regression trained for that specific bin; OTA = Orthopaedic Trauma Association fracture classification of level of complication or location of fracture.

**Table 2 sensors-24-05321-t002:** Selected features.

Feature	FS Model	VIF	Importance Score					
			DHA	DH4	DH3	DH2	DH1	DH0
OTA complication	Both	1.67	−3.20	4.59	−2.78	−1.76	−1.65	1.80
OTA location	Both	1.44	2.38	−5.89	2.79	1.48	0.95	−0.78
Mean of the medial sensor	DT	1.60	−1.25	3.74	−1.30	−1.23	−0.67	0.49
Day	Both	1.23	2.99	−0.37	0.25	1.24	2.32	−2.99
Variance of the medial sensor	DT	2.44	1.60	−1.68	0.86	1.46	1.04	−0.96
Heel to medial ratio	Lasso	5.19	−1.84	0.46	−0.32	0.03	−1.66	2.28
Mean of Diff_HF	Lasso	1.13	1.33	0.88	0.28	0.62	1.22	−1.46
Gender	Both	2.70	−0.79	2.41	−0.63	0.00	−0.48	0.83
Variance of the total waveform	Lasso	2.86	0.20	−3.02	0.55	−0.37	−0.08	−0.13
Lateral to the medial ratio	DT	4.81	−1.00	0.14	−1.11	−2.43	0.02	−0.04
Location of the maximum on Diff_HF	Lasso	3.55	−0.29	0.01	0.04	−0.12	−0.29	0.41

Note: FS = features selection; DT = decision tree; VIF = variance inflation factor, DH = bins based on days to healed; DHA = importance score from logistic regression trained to classify all DH bins; DH4-0 = importance score from logistic regression trained for that specific bin; OTA = Orthopaedic Trauma Association fracture classification of level of complication or location of fracture.

**Table 3 sensors-24-05321-t003:** Cross-validation.

Leave one patient out		Precision	Recall	F1-score	Accuracy
	Minimum	75.2%	74.4%	74.2%	74.4%
	Mean	76.7%	76.0%	75.9%	76.0%
	Maximum	78.9%	78.5%	78.1%	78.5%
10 fold cross-validation		Precision	Recall	F1-score	Accuracy
	Minimum	75.0%	74.7%	74.4%	74.7%
	Mean	76.1%	75.6%	75.4%	75.6%
	Maximum	77.5%	76.9%	76.7%	76.9%

## Data Availability

Data and code are available through the GitHub repository at: https://github.com/Garangatang/Multi_Logistic_Regression_Classifier/tree/main. (accessed on 16 August 2024)

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
