# Peer review of "Predicting the Healing of Lower Extremity Fractures Using Wearable Ground Reaction Force Sensors and Machine Learning"

_sensors, 2024, doi:10.3390/s24165321_

Round 1

Reviewer 1 Report

Comments and Suggestions for Authors

The paper presents a promising approach to predicting the healing of lower extremity fractures (LEFs) using wearable sensors and machine learning. The integration of wearable technology with machine learning to provide objective assessment and prediction of fracture healing is interesting and innovative. The study is well-structured, and the methodology is sound. The manuscript is relevant to the readers of Sensors. However, some places require further clarification and improvement:

(1)   Table 1 could be expanded to include a brief explanation of each metric for readers unfamiliar with the specific terms. And it seems that the line number 128, 129, 130, 131are embed in the table. Please correct the format.

(2)   The study's sample size is relatively small, consisting of only 25 patients. A larger, more diverse patient population would enhance the generalizability of the findings. Future studies might aim to include a broader demographic range to account for variations in healing patterns across different ages, genders, and ethnicities.

(3) Please consider integrating additional sensors, such as accelerometers, gyroscopes, and pressure sensors, to capture a broader range of gait metrics. This multi-sensor approach can provide a more comprehensive assessment of the patient's movement patterns and enhance the predictive power of the machine learning models. Some studies are already considering muti-sensors, for example, “Matijevich, Emily S., et al. "Combining wearable sensor signals, machine learning and biomechanics to estimate tibial bone force and damage during running." Human movement science 74 (2020): 102690”

(4)   The primary outcome measure, the "days to healed" metric, is based on periodic X-ray assessments. This measure may not capture the full spectrum of healing progress, particularly in terms of functional recovery and soft tissue healing. Please consider incorporating additional outcome measures, such as patient-reported outcomes and functional assessments, would provide a more holistic view of fracture healing.

Comments on the Quality of English Language

The manuscript is generally clear and easy to follow. The authors have done a good job of explaining complex technical concepts in a way that is accessible to a broad audience. 

Reviewer 2 Report

Comments and Suggestions for Authors

Methodology can be improved. Please include the following in the methods. Develop, Support Vector Classifier, Random Forest Classifier. Provide the comparative account, propose the best model based on the results. Include Optimization of hyperparameters of these models. Create Data pipelines, if this going to be a production level model and will be deployed on Cloud. Please include univariate and bivariate analysis in the data visualization.

Reviewer 3 Report

Comments and Suggestions for Authors
  • The main contribution of the research paper is the integration of wearable ground reaction force (GRF) sensors with machine learning (ML) to objectively monitor healing progression in lower extremity fracture rehabilitation.

Suggestions for Improving the Paper:

  1. Enhance model accuracy by refining the setup or using more frequent metrics than x-rays for determining healing [2].
  2. Consider extending the classification range beyond 30 days to reduce overlap and improve accuracy [2].
  3. Explore additional machine learning models or algorithms to compare and enhance predictive capabilities.
  4. Conduct a larger-scale study involving more participants to strengthen the generalizability of the findings.
  5. Investigate the impact of different types of lower extremity fractures on healing prediction accuracy.
  6. Include a detailed discussion on the practical implications of the research findings for clinicians and patients [1].
  7. Provide a more in-depth analysis of the feature selection process and its relevance to fracture healing prediction.
  8. Consider incorporating patient feedback and perspectives to further validate the effectiveness and usability of the proposed approach.
  9. Explore the potential integration of other wearable sensors or technologies to complement GRF data for a more comprehensive assessment.
  10. Discuss the scalability and feasibility of implementing the proposed system in real-world clinical settings for widespread adoption and impact.
  11.  Formal methods can be used to verify the correctness of smart contract code, which can help to prevent costly errors and security breaches. Therefore, it is important to discuss the use of formal methods in your paper.
  12. For this purpose, the authors may include the following interesting references (and others):
    a. https://ieeexplore.ieee.org/document/9970534

    b. https://ieeexplore.ieee.org/document/8328737

Round 2

Reviewer 3 Report

Comments and Suggestions for Authors

The authors considered my comments and suggestions. Good luck.